J Physiol 603.22 (2025) pp 7171–7188

7171

# Adeno-associated virus-based rescue of *Myo7a* expression restores hair-cell function and improves hearing thresholds in a *USH1B* mouse strain

Ana E. Amariutei[1] , Samuel Webb[1], Adam J. Carlton[1], Andrew O'Connor[1], Anna Underhill[1],
Jing-Yi Jeng[1], Sarah A. Hool[1], Alice Zanella[1], Matthew Hool[1], Marie-José Lecomte[2],
Stuart L. Johnson[1,3], Saaid Safieddine[2] and Walter Marcotti[1,3]

[1] *School of Biosciences, University of Sheffield, Sheffield, UK*
[2] *Université Paris Cité, Institut Pasteur, AP-HP, INSERM, CNRS, Fondation Pour l'Audition, Institut de l'Audition, IHU reConnect, Paris, France*
[3] *Neuroscience Institute, University of Sheffield, Sheffield, UK*

Handling Editors: Vaughan Macefield & Conny Kopp-Scheinpflug

The peer review history is available in the Supporting Information section of this article (https://doi.org/10.1113/JP289526#support-information-section).

Dr **Ana E. Amariutei** received her MSc and PhD in Biomedical Science at the University of Sheffield (UK) under the supervision of Professor Walter Marcotti. Her research focused on hair cell functional regeneration and gene therapy approaches for hearing loss. Now, as a consultant, Ana shapes clinical trial design, accelerates patient engagement and drives evidence generation. She partners with pharmaceutical companies, patient organizations and policy-makers to develop innovative care models and solutions that address unmet needs and improve patient health outcomes. Dr **Samuel Webb** is a postdoctoral researcher within the Hearing Research Group at the University of Sheffield. He earned his PhD in auditory neuroscience from Manchester Metropolitan University in 2019 and now focuses on uncovering the biological mechanisms underlying auditory function and dysfunction. His current research examines how ageing and noise exposure drive auditory decline, aiming to identify key mechanisms that can be targeted to improve long-term hearing health.

A. E. Amariutei and S. Webb contributed equally to this work.

**Abstract figure legend** *Shaker-1* mice, a murine model of Usher 1B syndrome, carry a mutation in the unconventional myosin MYO7A. This results in the progressive loss of the two shortest rows of stereocilia that house the mechano-electrical transducer (MET) channels in hair cells, leading to deafness shortly after hearing onset. Exogenous delivery of *Myo7a* to newborn *Shaker-1* mice using dual-adeno-associated virus (AAV) vectors substantially rescued inner hair cell stereocilia and mechanoelectrical transduction, resulting in improved hearing function. BK: large conductance $Ca^{2+}$ activated $K^+$ channel.

**Abstract** Mutations in *MYO7A*, the gene encoding the unconventional myosin 7a, cause hereditary deafness in mice and humans. In the cochlea, MYO7A is present in the sensory hair cells from embryonic stages of development, and plays a critical role in the development and maintenance of the mechanosensitive hair bundles composed of actin-rich stereocilia. *Shaker-1* mutant mice ($Myo7a^{Sh1/Sh1}$), the murine model of Usher 1B syndrome, exhibit a progressive loss of the stereocilia, subsequent degeneration of the sensory epithelium and ultimately profound deafness. In addition to the hair bundle defects, we found that the *shaker-1* mutation prevented both inner hair cells (IHCs) and outer hair cells (OHCs) from acquiring their fully mature basolateral current profile. Delivering exogenous *Myo7a* to newborn $Myo7a^{Sh1/Sh1}$ mice using dual-adeno-associated virus 8 (AAV8)-*Myo7a* or dual-AAV9-PhP.eB-*Myo7a*, which primarily target IHCs, led to a substantial rescue of their hair bundle structure. The rescued bundles regained their ability to generate mechano-electrical transducer (MET) currents in response to fluid jet displacement. Although the average MET current was smaller than in control IHCs, the normal resting open probability of the MET channel was fully restored. The IHCs of the treated cochlea also regained a mature basolateral membrane current profile. Functionally, rescue of the IHC structure and function, but not that of OHCs, leads to an average improvement of 20–30 dB in hearing thresholds across most frequencies. These results support dual AAV-induced gene replacement therapy as an effective strategy to recover hair-cell function in $Myo7a^{Sh1/Sh1}$ mice.

(Received 18 June 2025; accepted after revision 28 August 2025; first published online 27 September 2025)

**Corresponding author** W. Marcotti: School of Biosciences, University of Sheffield, Sheffield, S10 2TN, UK. Email: w.marcotti@sheffield.ac.uk

## Key points

- *Shaker-1* mutant mice ($Myo7a^{Sh1/Sh1}$), which carry a mutation in the unconventional myosin MYO7A and are the murine model of Usher 1B syndrome, become profoundly deaf at 1 month of age or soon after.
- In the mammalian cochlea, MYO7A is expressed in the hair cells, including within their actin-rich stereociliary bundles.
- We show that hair cells of $Myo7a^{Sh1/Sh1}$ mice progressively lose their transducing stereocilia and mechanoelectrical transduction, and fail to acquire their fully mature basolateral current profile.
- Delivering exogenous *Myo7a* to newborn $Myo7a^{Sh1/Sh1}$ mice using dual-adeno-associated virus (AAVs) led to a substantial rescue of the bundle structure and function of inner hair cells, including mechanoelectrical transduction.
- This functional rescue led to a 20–30 dB improvement in hearing thresholds across most frequencies.
- These results support dual AAV-induced gene replacement therapy as an effective strategy to recover the hair-cell function in $Myo7a^{Sh1/Sh1}$ mice.

## Introduction

*MYO7A* was the first deafness gene identified in humans causing both syndromic (*USHER 1B*) and non-syndromic recessive deafness (Liu et al., 1997; Weil et al., 1995, 1997). Mice homozygous for *Myo7a* mutations exhibit the characteristic *shaker-1* phenotype, which includes deafness, hyperactivity, head-tossing and circling (Gibson

et al., 1995; Lord & Gates, 1929). *Myo7a* encodes the unconventional myosin MYO7A, which is part of a class of motor proteins that couple ATP hydrolysis with mechanical force, allowing them to move along actin-based filaments (Goode et al., 2000). In the mammalian cochlea, MYO7A is expressed in the hair cells, including within their stereociliary bundles (e.g. Hasson et al., 1995; Kolla et al., 2020; Rzadzinska et al., 2004; Scheffer et al., 2015; Underhill et al., 2025). Stereocilia are actin-rich microvilli-like structures required for the transduction of acoustic stimuli into a sensory receptor potential (Fettiplace & Kim, 2014). MYO7A has been implicated in several mechanisms, most of which are associated with the development of the hair bundles (Ballesteros et al., 2022; Moreland & Bird, 2022; Peng et al., 2009; Rzadzinska et al., 2009; Senften et al., 2006). More recently, MYO7A has been shown to be essential for maintaining the structural integrity of the mature hair bundles in post-hearing mice (Underhill et al., 2025).

Several recessive mutations have been identified in *Myo7a*, all of which cause defects in the cochlear hair cells and lead to deafness, though the onset and progression of the dysfunction vary (Libby & Steel, 2001; Lord & Gates, 1929; Self et al., 1998). Mutation in the *Myo7a⁴⁶²⁶ˢᴮ* allele results in a stop codon within the head domain of MYO7A, resulting in the absence of protein expression in the cochlea and severe hair cell abnormalities (Hasson et al., 1997; Kros et al., 2002; Mburu et al., 1997). The *Myo7a⁸¹⁶ˢᴮ* mutation, which results in a 10-amino-acid deletion in the MYO7A motor head core (Gibson et al., 1995; Mburu et al., 1997), also causes severe hair bundle abnormalities from early stages of development and profound deafness. The *Myo7a⁶ᴶ* mutation is an arginine to proline missense mutation that is also located in the core of the motor domain and is associated with a rapid progression of stereocilia dysfunction and deafness. Both *Myo7a⁸¹⁶ˢᴮ* and *Myo7a⁶ᴶ* mutations were predicted to have severe effects on protein stability (80–95% reduction: Hasson et al., 1997) and function (Self et al., 1998). In contrast to the mutations described above, the original *shaker-1* mutant mice (*Myo7aˢʰ¹/ˢʰ¹*), which also have an arginine-to-proline missense mutation located in a poorly conserved surface loop of the motor head (Gibson et al., 1995; Mburu et al., 1997), exhibit normal MYO7A expression in the hair cells. The hair bundles in these mice appear to develop normally, at least initially, with residual hearing function in very young adult mice (Self et al., 1998; Shnerson et al., 1983).

Adeno-associated virus (AAV)-mediated gene replacement therapy has successfully recovered hearing in mouse models with mutations in several genes required for hair cell function, including *Otof* (Akil et al., 2019; Al-Moyed et al., 2019), *Tmc1* (Askew et al., 2015; Nist-Lund et al., 2019) and *Vglut3* (Akil et al., 2012; Zhao et al., 2022). However, recovering cochlear function through AAV-based replacement therapy has proven more challenging for genes that are critical during early stages of cochlear development (Amariutei et al., 2023). Indeed, a recent study aimed at reverting the cochlear phenotype in *Myo7a⁴⁶²⁶ˢᴮ/⁴⁶²⁶ˢᴮ* mice, characterized by severe hair bundle defects from early ages, was unable to rescue the hearing phenotype using dual-AAV vector-mediated expression of MYO7A in the inner ear (Lau et al., 2023). In the present study, we use *shaker-1* mice (*Myo7aˢʰ¹/ˢʰ¹*), which carry a mutation orthologous to that causing *USH1B* in humans (Weil et al., 1995), to test the capability of the dual-AAV to rescue hair cell function and hearing. We show that the exogenous delivery of *Myo7a* using AAV8 or AAV9-PhP.eB vectors was able to rescue several morphological and functional properties of IHCs, leading to an improvement in hearing thresholds of about 30 dB across most tested frequencies.

## Methods

### Ethical approval

Animal experimental work was licensed by the UK Home Office under the Animals (Scientific Procedures) Act 1986 (PCC8E5E93 and PP1481074) and was approved by the University of Sheffield Ethical Review Committee (180 626_Mar). Mice had free access to food and water and were on a 12 h light/dark cycle. Experiments were performed using mice carrying the spontaneous original *shaker-1* mutation (*Myo7aˢʰ¹*), obtained from the MRC Mary Lyon Centre (Harwell Campus, UK) and maintained on the original background (85% CBA and 15% mixed) (Gibson et al., 1995). Littermate mice of either sex were used for experiments.

For *ex vivo* experiments mice were killed by cervical dislocation followed by decapitation. For *in vivo* measurement of auditory brainstem responses (ABRs), mice were anaesthetized using intraperitoneal injection of ketamine (100 mg/kg body weight, Fort Dodge Animal Health, Fort Dodge, IA, USA) and xylazine (10 mg/kg, Rompun 2%, Bayer HealthCare LLC, Whippany, NY, USA). Mice were placed in a soundproof chamber for *in vivo* experiments following the loss of the retraction reflex with a toe pinch. After ABR recordings were carried out, mice were either killed by cervical dislocation or recovered with an intraperitoneal injection of atipamezole (1 mg/kg), permitting recovery from anaesthesia. For *in vivo* gene delivery, mice were anaesthetized with isoflurane (2.5%) delivered in 100% oxygen at a flow rate of 0.8 L/min. Mice under recovery from anaesthesia were returned to their cage, placed on a thermal mat, and monitored over the following 2–5 h until normal behaviour resumed.

All animal experiments performed in this study comply with *The Journal of Physiology*'s policies regarding

animal experiments (https://physoc.onlinelibrary.wiley.com/hub/animal-experiments).

## Tissue preparation

Both male and female mice were used for *ex vivo* experiments. Following cervical dislocation, the cochlea was dissected out and placed in a chamber containing an extracellular solution composed of (in mM): 135 NaCl, 5.8 KCl, 1.3 $CaCl_2$, 0.9 $MgCl_2$, 0.7 $NaH_2PO_4$, 5.6 D-glucose and 10 Hepes-NaOH. Amino acids, vitamins and sodium pyruvate (2 mM) were added from concentrates (Thermo Fisher Scientific, UK). The final pH was 7.48 and osmolality ∼308 mOsm/kg. Following the dissection, the apical coil of the cochlear sensory epithelium was transferred to a microscope chamber and immobilized via a nylon mesh attached to a stainless-steel ring. The epithelia were continuously perfused with the above extracellular solution using a peristaltic pump (Cole-Palmer, UK) connected to the microscope chamber, which was then mounted on the stage of an upright microscope (Olympus BX51, Tokyo, Japan; Leica DMLFS, Wetzlar, Germany). Microscopes were equipped with Nomarski differential interference contrast (DIC) optics, either 60× or 64× water immersion objective and 15× eyepieces.

## Auditory brainstem responses

Anaesthetized male or female mice were placed on a heated mat (37°C) inside a soundproof chamber (MAC-3 acoustic chamber, IAC Acoustic, Chandler's Ford, UK). The mouse's pinna was positioned at 10 cm from the loudspeaker (MF1-S, Multi Field Speaker, Tucker-Davis Technologies, Alachua, FL, USA), which was calibrated daily with a low-noise microphone probe system (ER10B+, Etymotic, USA). One subdermal electrode was positioned half-way between the two pinna on the vertex of the cranium (active electrode) and the other two electrodes were placed under the skin behind the pinna of each ear (reference and ground electrode) as previously described (Carlton et al., 2024). Experiments were performed using customized software (Ingham et al., 2011) driving an RZ6 auditory processor (Tucker-Davis Technologies). Response thresholds were estimated from the resulting ABR waveform, defined as the lower sound level at which any recognizable feature of the waveform was visible. Responses were recorded for pure tones of frequencies at 3, 6, 12, 18, 24, 30 and 36 kHz as well as broadband white noise clicks. Stimulus sound pressure levels were up to 95 or 120 dB SPL, presented in steps of 5 dB SPL (average of 256 repetitions). Tone bursts were 5 ms in duration with a 1 ms on/off ramp time presented at a rate of 42.6/s.

## Whole-cell electrophysiology

Patch clamp recordings were performed at room temperature (20–24°C) using an Optopatch amplifier (Cairn Research Ltd, Faversham, UK) as previously described (Carlton et al., 2023). Patch pipettes were pulled from soda glass capillaries, which had a typical resistance in extracellular solution of 2–3 MΩ. The patch pipette intracellular solution contained (in mM): 131 KCl, 3 $MgCl_2$, 1 EGTA-KOH, 5 $Na_2ATP$, 5 Hepes-KOH, 10 Na-phosphocreatine (final pH: 7.28; 294 mOsm/kg). Data acquisition was controlled by pClamp software using a Digidata 1440A (Molecular Devices, Sunnyvale, CA, USA). To reduce the electrode capacitance, patch electrodes were coated with surf wax (Mr Zoggs SexWax, USA). Recordings were low-pass filtered at 2.5 kHz (8-pole Bessel), sampled at 5 kHz and stored on a computer for off-line analysis (Clampfit, Molecular Devices; Origin 2023: OriginLab, Northampton, MA, USA). Membrane potentials under voltage-clamp conditions were corrected off-line for the residual series resistance $R_s$, which was normally compensated for by 80%, and the liquid junction potential (LJP) of −4 mV, which was measured between electrode and bath solutions.

To investigate the biophysical characteristics of the mechanoelectrical transducer (MET) current, hair bundles were displaced using a fluid-jet system from a glass pipette driven by a 25 mm diameter piezoelectric disc (Carlton et al., 2021; Corns et al., 2014; Underhill et al., 2025). The fluid jet pipette tip was positioned near the hair bundles to elicit a maximal MET and contained the same extracellular solution mentioned above. Mechanical stimuli were applied as 50 Hz sinusoids (filtered at 1 kHz, 8-pole Bessel). Prior to positioning of the fluid jet near the hair bundles, any steady-state pressure was removed by monitoring the movement of debris in front of the pipette.

## AAV production

Due to size constrains, the full-length coding sequence of the murine *Myo7a* cDNA (NM_0 011 00395.1) was split into two fragments: a 5′ fragment covering nucleotides 1–3108 and a 3′ fragment covering nucleotides 3109–6648. The 5′ construct (p0101_NterMYO7a) included the 5′ part of the *Myo7a* cDNA (encoding amino acids 1–1036) as well as a splice donor (SD) site. The 3′ construct (p0101_CterMyo7a) encompassed the 3′ part of the *Myo7a* cDNA (encoding amino acids 1037–2215) and a splice acceptor (SA) site. The SA and SD sites enabled the reconstruction of the full-length transcript via trans-splicing within the transduced cells. All these fragments were synthesized by GenScript (Piscataway, NJ, USA). Both constructs, sharing the alkaline phosphatase recombinogenic

bridging sequence (AP), were inserted into a modified version of the pAAV.CMV.PI.EGFP.WPRE.bGH vector plasmid (Addgene, Cambridge, MA, USA), in which the EGFP coding sequence was replaced by the *Myo7a* fragment. This resulted in the generation of a pair of recombinant vectors referred to as AAV-MYO7A-N-term and AAV-MYO7A-C-term. The recombinant vectors were packaged in-house into the AAV2/8 or AAV9-PhP.eB capsids. AAV titres were expressed in viral genomes per millilitre (vg/mL) as determined by a fluorometric assay: AAV8-MYO7A-N-term ($1.0 \times 10^{13}$ vg/mL); AAV8-MYO7A-C-term ($6.3 \times 10^{12}$ vg/mL); AAV9-PhP.eB-MYO7A-N-term ($1.5 \times 10^{13}$ vg/mL); AAV9-PhP.eB-MYO7A-C-term ($6.8 \times 10^{12}$ vg/mL); AAV8-GFP ($2.8 \times 10^{13}$ vg/mL); AAV9-PhP.eB-GFP ($2.2 \times 10^{13}$ vg/mL). Final concentrated AAV vector stocks were stored in phosphate-buffered saline (PBS) with $MgCl_2$ (1 mM) and KCl (2.5 mM) at −70°C.

### AAV gene delivery in mice

The surgical protocol used for AAV injection into the cochlea of P0–P1 *Myo7a*$^{Sh1}$ mice was performed under general anaesthesia. The left or right ear was accessed via an incision just below the pinna as previously described (Jeng et al., 2022; O'Connor et al., 2024). When the round window membrane (RWM) was identified, it was gently punctured with a borosilicate pipette. This was followed by the injection of the AAV into the cochlea (pressure controlled by mouth) of 1 or 2 μL of the AAVs. For injections involving dual-AAV constructs, AAV8-*Myo7a* ($1.2 \times 10^{12}$ vg/mL) or AAV9-PhP.eB-*Myo7a* ($6.8 \times 10^{12}$ vg/mL) were used, prepared by combining equal volumes of the corresponding N-terminal and C-terminal vectors. Following the injection, the pipette was retracted from the RWM and the wound was closed with veterinarian glue.

### Scanning electron microscopy

After dissecting out the inner ear from the mouse, the cochlea was gently perfused with fixative for 1—2 min through the round window using a 10 μL pipette tip. A small hole in the apical portion of cochlear bone was made prior to perfusion to allow the fixative to flow out from the cochlea. The fixative contained 2.5% (v/v) glutaraldehyde in 0.1 M sodium cacodylate buffer plus 2 mM $CaCl_2$ (pH 7.4). The inner ears were then immersed in the above fixative and placed on a rotating shaker for 2 h at room temperature. After the fixation, the cochleae were washed in the same cacodylate buffer and the organ of Corti was exposed by removing the bone from the apical coil of the cochlea. Samples were then immersed in 1% osmium tetroxide in 0.1 M cacodylate buffer

for 1 h. For osmium impregnation, which avoids gold coating, cochleae were incubated in solutions of saturated aqueous thiocarbohydrazide (20 min) alternating with 1% osmium tetroxide in buffer (2 h) twice (the OTOTO technique: Furness & Hackney, 1986). The cochleae were then dehydrated through an ethanol series and critical point dried using $CO_2$ as the transitional fluid (Leica EM CPD300) and mounted on specimen stubs using conductive silver paint (Agar Scientific, Stansted, UK). The apical coil of the sensory epithelium was examined at 10 kV using a Tescan Vega3 LMU scanning electron microscope (Cryo-Electron Microscopy facility, University of Sheffield) or an FEI Inspect F scanning electron microscope (Sorby Centre for Electron Microscopy, University of Sheffield). At least three mice were processed for each genotype. Images were taken from the same region (around 12 kHz) used for the electrophysiological recordings.

### Immunofluorescence microscopy

For pre-hearing mice, the inner ear was dissected out and immersed for 20 min at room temperature in a solution containing 4% paraformaldehyde in PBS (pH 7.4). For adult mice, the inner ear was initially gently perfused with the above solution for 1–2 min through the round window prior to the 20 min fixation described for pre-hearing mice. Following fixation, the inner ears were then washed three times in PBS for 10 min and the sensory epithelia dissected out using fine forceps and incubated in PBS supplemented with 5% normal goat or horse serum and 0.5% Triton X-100 for 1 h at room temperature. The samples were immunolabelled with primary antibodies overnight at 37°C, washed three times with PBS and incubated with the secondary antibodies for 1 h at 37°C. Antibodies were prepared in 1% serum and 0.5% Triton X-100 in PBS. Primary antibodies were as follows: rabbit IgG anti-myosin 7a (1:500, Proteus Biosciences, #25-6790; Ramona, CA, USA), mouse IgG1 anti-BK (1:500, Antibodies Incorporated, 75–408; Davis, CA, USA), mouse IgG2a anti-GluR2 (1:200, Millipore, MAB397; Billerica, MA, USA) and mouse IgG1 anti-CtBP2 (1:500, BD, 612 044; Franklin Lakes, NJ, USA). F-actin was stained with Texas Red-X phalloidin (1:1000, ThermoFisher, T7471;.Waltham, MA, USA) within the secondary antibody solution. Secondary antibodies were species appropriate Alexa Fluor or Northern Lights. Samples were mounted in Vectashield (H-1000). The images from the apical cochlear region (around 12 kHz) were captured with Nikon A1 or ZEISS LSM980 Airyscan confocal microscopes (Wolfson Light Microscope Facility at the University of Sheffield). Image stacks were processed with Fiji ImageJ software. At least three mice for each genotype were used for each experiment.

## Statistical analysis

Statistical comparisons were made by Student's *t* test or Mann–Whitney *U* test (when a normal distribution could not be assumed). For multiple comparisons, one-way ANOVA followed by a suitable post test was used for normally distributed data, otherwise Kruskal–Wallis with Dunn's post test was used. $P < 0.05$ was selected as the criterion for statistical significance. Average values are quoted in the text and figures as means ± SD Animals of either sex were randomly assigned to the different experimental groups. No statistical methods were used to define sample size, which was defined based on similar previously published work from our laboratory. Animals were taken from several cages and breeding pairs over a period of several months.

## Results

### Missense mutation in the *shaker-1* gene disrupts the hair bundle morphology and causes hearing loss

ABRs, which reflect the electrical activity of the afferent spiral ganglion neurons and downstream auditory pathway, were used to investigate the hearing function of $Myo7a^{Sh1}$ mice. ABR thresholds for click sound stimuli were absent at P26–P38 in $Myo7a^{Sh1/Sh1}$ compared to $Myo7a^{Sh1/+}$ mice ($P < 0.0001$, Fig. 1*A*). When frequency-specific pure tone burst stimuli were used, ABR thresholds were also absent or highly elevated in $Myo7a^{Sh1/Sh1}$ compared to control ($Myo7a^{Sh1/+}$) mice ($P < 0.0001$, two-way ANOVA, Fig. 1*B*). Hearing loss in $Myo7a^{Sh1/Sh1}$ mice was associated with the lack of the shorter second and third rows of stereocilia in both inner hair cells (IHCs) and outer hair cells (OHCs) (Fig. 1*C* and *D*). Since the shorter stereocilia rows are those with the MET channels at their tip (Beurg et al., 2009), the loss of channels is likely to be the primary cause of deafness in *shaker-1* mice. Although previous studies have indicated that the hair bundles of both hair cell types develop normally, at least during early postnatal ages (Kros et al., 2002; Self et al., 1998), we found that the loss of stereocilia was already evident at P10 (Fig. 1*E* and *F*), which is before the onset of hearing (about P12–P13: Shnerson & Pujol, 1982).

### The basolateral membrane profile of OHCs and IHCs in *Myo7a^{Sh1}* mice

Since a functional MET apparatus has been shown to be essential for maintaining the biophysical identity of adult IHCs and OHCs (Corns et al., 2018; O'Connor et al., 2024), we investigated whether a similar functional regulation was also present in $Myo7a^{Sh1/Sh1}$ mice that lack the transducing stereocilia. Basolateral membrane currents were recorded from immature and mature hair cells by applying a series of voltage steps from −124 mV to more positive potentials in 10 mV nominal increments from the holding potential of −84 mV.

In immature P5 OHCs, we found that their resting membrane potential and current profile, which is defined by the expression of a delayed outward rectifying $K^+$ current ($I_K$) and an inward rectifying $K^+$ current ($I_{K1}$), was indistinguishable between control and $Myo7a^{Sh1/Sh1}$ mice (Fig. 2*A*–*E*). This indicates that the initial development of the MET apparatus is likely to be unaffected by the *shaker-1* mutation, as previously suggested (Self et al., 1998). The onset of functional maturation in apical OHCs occurs at around P7–P8 marked by the down-regulation of ion channels that carry the immature-type currents (e.g. P5: Fig. 2*A* and *B*), the emergence of the negatively activating $K^+$ current $I_{K,n}$ carried by KCNQ4 channels (Kubisch et al., 1999) required for setting the resting membrane potential (Marcotti & Kros, 1999) and the acquisition of electromotility (Abe et al., 2007). Although mature OHCs from P17–P18 $Myo7a^{Sh1/Sh1}$ mice successfully down-regulated their immature-type currents, they failed to fully upregulate $I_{K,n}$ (Fig. 2*F*–*I*), resulting in a significantly more depolarized resting membrane potential compared to control cells (Fig. 2*J*). These data suggest that a defect in the MET apparatus of OHCs from $Myo7a^{Sh1/Sh1}$ mice is likely to occur at or following the onset of maturation during the second post-natal week.

IHC functional maturation occurs at around the onset of hearing (P12: Kros et al., 1998; Marcotti et al., 2003), by which time OHCs have already acquired mature-like characteristics (Marcotti & Kros, 1999). We found that the size of both $I_K$ and $I_{K1}$ in IHCs was already significantly reduced in P10 $Myo7a^{Sh1/Sh1}$ mice compared to control littermates (Fig. 3*A*–*D*), a developmental stage when hair bundle defects are already apparent (Fig. 1*C* and *D*). A previous study showed that IHC ribbon synapses from mice lacking the MET current fail to develop normally, with their number appearing normal at P2 but becoming markedly elevated by P7, indicating defects in their developmental refinement (Lee et al. 2021). We made similar observations in the *shaker-1* mutant, with the number of pre-synaptic ribbons (CtBP2 puncta), post-synaptic AMPA glutamate receptors (GluR2 puncta) and their colocalization being significantly increased in $Myo7a^{Sh1/Sh1}$ mice compared to control littermates (Fig. 3*E* and *F*).

Mature IHCs express a rapidly activating, large conductance $Ca^{2+}$-activated $K^+$ current carried by BK channels, named $I_{K,f}$ (Kros et al., 1998; Lingle et al., 2019; Marcotti et al., 2004; Thurm et al., 2005), which was evident in control cells but largely reduced in $Myo7a^{Sh1/Sh1}$ mice (Fig. 4*A*–*C*). In addition, post-hearing IHCs express, like OHCs, $I_{K,n}$ (Kros et al., 1998; Marcotti et al., 2003;

Oliver et al., 2003), which was also largely reduced in *Myo7a*$^{Sh1/Sh1}$ mice (Fig. 4*D* and *E*). These results highlight that the mutation in *Myo7a* present in *shaker-1* mice disrupts the normal acquisition of the biophysical characteristics of mature hair cells.

### AAV-mediated *Myo7a* replacement in new born mice improves hearing thresholds and repairs IHC morphological and biophysical defects in *Myo7a*$^{Sh1}$ mice

To determine whether hearing loss caused by the missense mutation *Myo7a*$^{Sh1}$ can be reversed by AAV-based gene replacement therapy, we generated dual-AAV8-*Myo7a* and dual-AAV9-PhP.eB-*Myo7a* vectors. Since similar results were obtained with both AAV serotypes, the data were pooled for analysis. We first investigated whether both the AAV8 and AAV9-PhP.eB serotypes were effective in transducing hair cells in the inner ear of the *shaker-1* mice by assessing their transduction efficiency using GFP-only reporter constructs. We injected AAV8-GFP or AAV9-PhP.eB-GFP into the perilymphatic space via the RWM in P1–P3 C57BL/6N mice and quantified GFP-positive hair cells (Fig. 5*A*). We quantified only the number of GFP-positive IHCs (Fig. 5*B*), as OHCs were rarely transduced by either AAV vector (Fig. 5*A*).

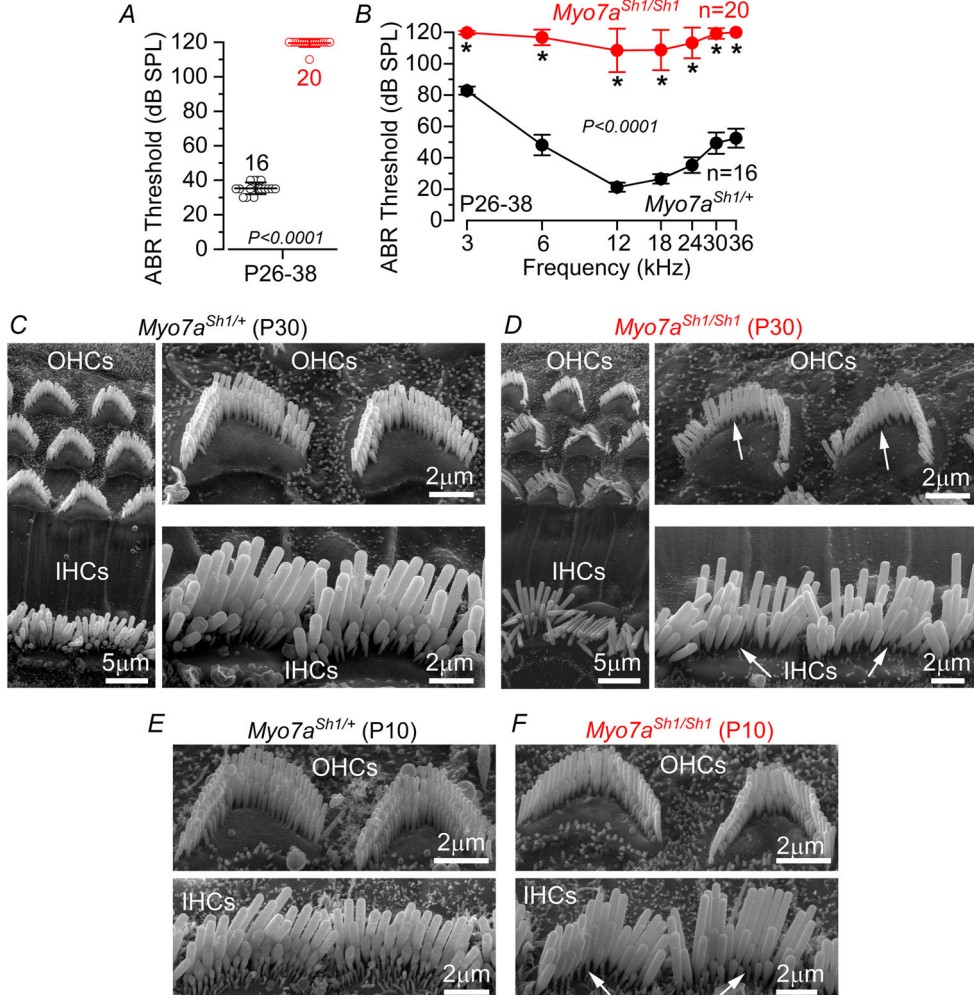

**Figure 1. Hearing function and hair bundle morphology in *Myo7a*$^{Sh1}$ mice**
*A*, average auditory brainstem response (ABR) thresholds for click stimuli recorded from control *Myo7a*$^{Sh1/+}$ (black) and *Myo7a*$^{Sh1/Sh1}$ (red) mice at postnatal day 26–38 (P26–38). Click thresholds were significantly elevated in *Myo7a*$^{Sh1/Sh1}$ compared to control mice ($P < 0.0001$ Mann–Whitney *U*-test). *B*, ABR thresholds for frequency-specific pure tone burst stimuli at 3, 6, 12, 18, 24, 30 and 36 kHz recorded from controls and littermate *Myo7a*$^{Sh1/Sh1}$ mice at P26–38 (*$P < 0.0001$ from Šidák's post test, two-way ANOVA). In both *A* and *B*, the numbers of mice tested for each genotype are shown next to the symbols and values are reported as mean ± SD. *C–F*, example of scanning electron microscope images showing the hair bundles of IHCs and OHCs from the apical coil of the cochlea at P30 (*C* and *D*) and P10 (*E* and *F*) of control (*C* and *E*) and *Myo7a*$^{Sh1/Sh1}$ (*D* and *F*) mice. Arrows point to the missing stereocilia in the second and/or third row of the hair cells from *Myo7a*$^{Sh1/Sh1}$ mice.

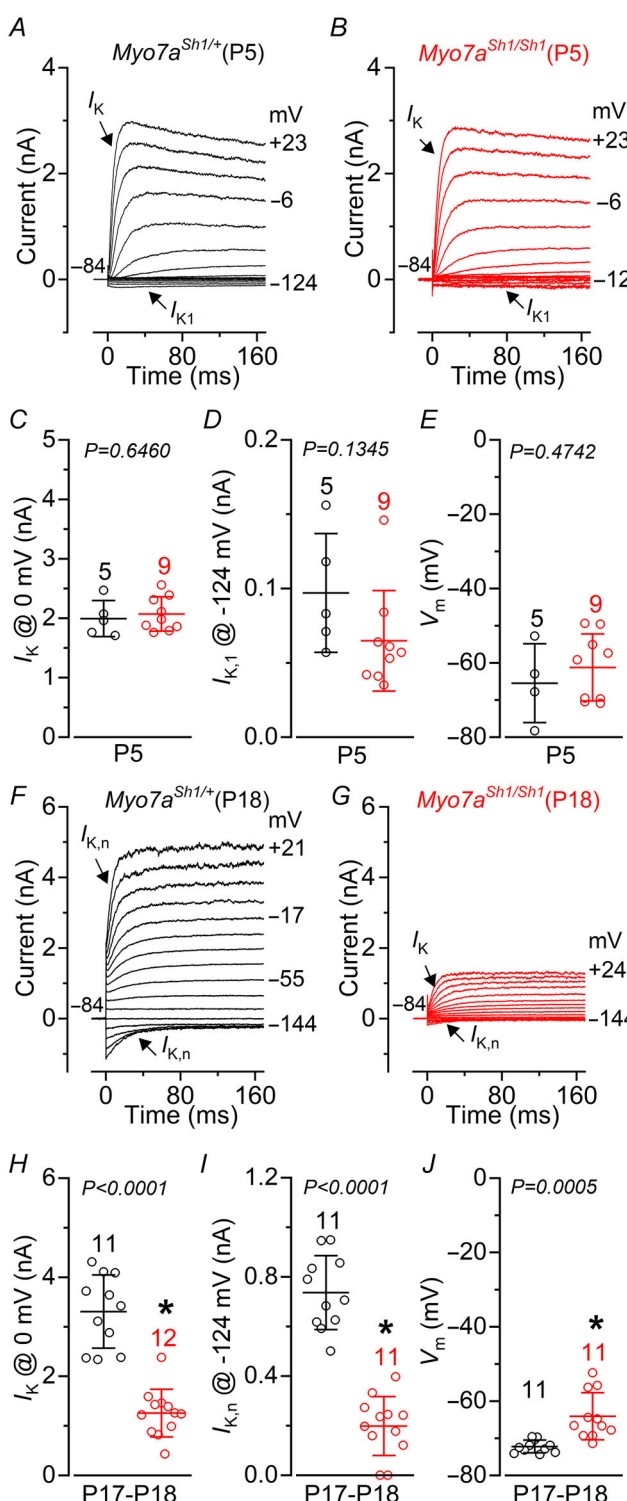

currents in both genotypes. *C* and *D*, average size of the peak total outward $K^+$ current measured at 0 mV $I_K$ (*C*) and $I_{K,1}$ measured at −124 mV (*D*) in OHCs from control (black) and *Myo7a*$^{Sh1/Sh1}$ (red) mice. *E*, average resting membrane potential measured in the OHCs from both genotypes. *F* and *G*, current responses from mature P18 OHCs of control (*F*) and *Myo7a*$^{Sh1/Sh1}$ (*G*) mice obtained as described in *A* and *B*. *H* and *I*, average size of the peak total outward $K^+$ current measured at 0 mV $I_K$ (*H*) and $I_{K,n}$, which was measured as the difference between the peak and steady-state of the deactivating inward current at −124 mV (*I*) in OHCs from control (black) and *Myo7a*$^{Sh1/Sh1}$ (red) mice. *J*, average resting membrane potential measured in the OHCs from both genotypes. Data in *C*–*E* and *H*–*J* are plotted as mean ± SD. Single cell value recordings (open symbols) are plotted behind the average bars. Statistical tests shown are obtained using a *t* test. Number of IHCs investigated is shown above the average data points.

Since the missense mutation in *Myo7a*$^{Sh1}$ does not prevent the expression of MYO7A in hair cells, it was not possible to identify which hair cells had been transduced by the AAV-*Myo7a* (Fig. 6*A*). Considering that the main phenotype of the *shaker-1* mutation is the loss of the two transducing rows of stereocilia (rows 1 and 2: see Fig. 1), we assessed the degree of IHC recovery by quantifying the number of rows in control, *Myo7a*$^{Sh1/Sh1}$ and *Myo7a*$^{Sh1/Sh1}$ mice injected with AAV-*Myo7a* (Fig. 6*B*–*E*). While the hair bundles of P37 control IHCs were all formed by three rows of stereocilia (Fig. 6*B* and *E*), those from littermate *Myo7a*$^{Sh1/Sh1}$ mice consisted primarily of a single row of stereocilia (Fig. 6*C* and *E*). *Myo7a*$^{Sh1/Sh1}$ mice injected with AAV-*Myo7a* showed a significantly increased number of IHCs with two and three rows of stereocilia compared to *Myo7a*$^{Sh1/Sh1}$ mice (Fig. 6*D* and *E*).

Considering the much-improved hair bundle morphology following AAV-*Myo7a* injection, we investigated whether this led to a recovery in the biophysical properties of IHCs. MET currents were investigated from mature IHCs in response to hair bundle displacement using a 50 Hz sinusoidal force stimulus from a piezo-driven fluid jet (Carlton et al., 2021; Corns et al., 2018; Underhill et al., 2025). Hair bundle displacement in the excitatory direction (i.e. towards the taller stereocilia) elicited an inward MET current at the holding potential of −84 mV in control P40 mice (Fig. 7*A* and *D*). The same approach applied to the IHCs of *Myo7a*$^{Sh1/Sh1}$ mice failed to elicit any MET current (Fig. 7*B* and *D*), which agrees with most of these cells having lost the two transducing rows of stereocilia (Fig. 6). However, an MET current, although smaller than that in control IHCs, was present in *Myo7a*$^{Sh1/Sh1}$ mice injected with AAV-*Myo7a* (Fig. 7*C* and *D*), consistent with most of them having regained one or both transducing rows of stereocilia (Fig. 6). The resting open probability ($P_o$) of the MET channel, which is responsible for the current flowing in the absence of mechanical stimulation, was also present

**Figure 2. OHCs from *Myo7a*$^{Sh1/Sh1}$ mice fail to develop their characteristic mature current profile**

*A* and *B*, current responses from apical coil OHCs of immature P5 control *Myo7a*$^{Sh1/+}$ (*A*) and *Myo7a*$^{Sh1/Sh1}$ (*B*) mice. Current recordings were elicited by using depolarizing and hyperpolarizing voltage steps (10 mV increments) from the holding potential of −84 mV to the various test potentials shown by some of the traces. Note the similar size and time-course of the outward and inward $K^+$

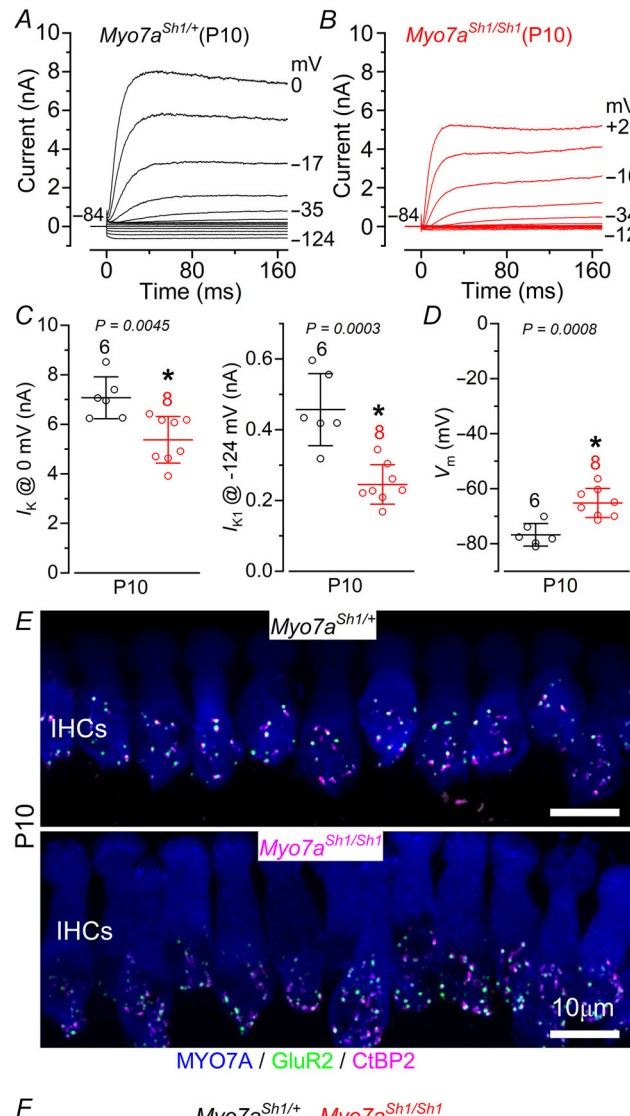

apical cochlear region of *Myo7a*$^{Sh1/+}$ (upper panel) and *Myo7a*$^{Sh1/Sh1}$ (lower panel) P10 mice. Cochleae were immunostained using antibodies against CtBP2 (ribbon synaptic marker: magenta) and GluR2 (postsynaptic marker: green). Myosin 7a (MYO7A) was used as the IHC marker (blue). *F*, number of GluR2 (left panel), CtBP2 (middle) and colocalized GluR2-CtBP2 puncta (right) from *Myo7a*$^{Sh1/+}$ and *Myo7a*$^{Sh1/Sh1}$ P10 mice. Statistical analysis was performed using the Mann–Whitney *U*-test. Data are plotted as mean values and individual counts (open symbols) are also shown. Data are plotted as mean ± SD.

in *Myo7a*$^{Sh1/Sh1}$ mice injected with AAV-*Myo7a*, the size of which was comparable to that of the IHCs from control mice (Fig. 7*E*). In addition, IHCs from *Myo7a*$^{Sh1/Sh1}$ mice injected with AAV-*Myo7a* expressed the rapidly activating $I_{K,f}$ to a level comparable to that recorded in the IHCs of control mice (Fig. 7*F–J*).

Hearing function in these mice was then investigated at P25–P46 with ABR recordings, as described in Fig. 1. We found that ABR thresholds in mice that underwent the surgical procedure and delivery of AAV-GFP at P1–P3 were comparable to those of untreated control mice ($P = 0.9805$, two-way ANOVA, Fig. 8*A*). We then delivered the dual AAV vector carrying *Myo7a* into the cochlea via the RWM of P0–P1 control *Myo7a*$^{Sh1/+}$ and *Myo7a*$^{Sh1/Sh1}$ mice. Functional recovery was assessed between P26 and P38. ABR thresholds in response to pure tone burst stimuli differed significantly across experimental groups ($P < 0.0001$, two-way ANOVA, Fig. 8*B*). Notably, *Myo7a*$^{Sh1/Sh1}$ mice injected with either of the dual AAVs-*Myo7a* showed significantly improved ABR thresholds compared to un-injected *Myo7a*$^{Sh1/Sh1}$ mice ($P < 0.0001$, Tukey's post test, two-way ANOVA, Fig. 8*B*). Over the 12—24 kHz region, there was on average about 20—30 dB improvement in ABR threshold in AAV-*Myo7a* injected *Myo7a*$^{Sh1/Sh1}$ mice compared to *Myo7a*$^{Sh1/Sh1}$ mice (Fig. 8*B*). These results are particularly encouraging considering that both AAV vectors almost exclusively transduce IHCs (Fig. 5*A* and *B*).

## Discussion

The acquisition of hearing function requires the correct development of the stereociliary bundles on both IHCs and OHCs, which are responsible for the mechano-electrical transduction of acoustic stimuli into an electrical signal. This is a tightly regulated developmental process requiring the interaction of many proteins, including the unconventional myosins (Barr-Gillespie, 2015; McGrath et al., 2017; Park & Bird, 2023). The absence of functional MYO7A has been shown to cause severe morphological defects in the stereociliary bundles beginning from pre-hearing stages, albeit with different temporal progression depending on the mutation (Self

**Figure 3. *Shaker-1* mutation affects the IHC membrane current of both pre- and post-hearing ages**
*A* and *B*, current responses from IHCs of control *Myo7a*$^{Sh1/+}$ (*A*) and *Myo7a*$^{Sh1/Sh1}$ (*B*) P10 mice. Current recordings were elicited as described in Fig. 1. *C* and *D*, average size of the characteristic K$^+$ currents present in immature IHCs: outward delayed rectifier $I_K$, which was measured at 0 mV (*C*) (Marcotti et al., 2003), and the inward rectifier $I_{K,1}$, which was measured at –124 mV(*D*) (Marcotti et al., 1999) from control (black) and *Myo7a*$^{Sh1/Sh1}$ (red) P1 mice. Data in *C* and *D* are plotted as mean ± SD. Single cell value recordings (open symbols) are plotted behind the average data. Statistical tests shown are obtained using a *t* test. Number of IHCs investigated is shown above the average data points. *E*, maximum intensity projections of confocal z-stacks of IHCs taken from the

et al., 1998). Here we found that a missense mutation in the *shaker-1* gene ($Myo7a^{Sh1}$) led to hair bundle defects evident as early as P10, and by 1 month of age almost all hair cells had only one row of stereocilia. We also showed that both sensory hair cell types failed to acquire a fully mature basolateral electrophysiological profile. Exogenous delivery of *Myo7a* using dual-AAV vectors

in $Myo7a^{Sh1/Sh1}$ P0–P2 pups *in vivo* was able to partially restore the staircase structure of the hair bundles and the MET current in adult IHCs, which were the main target of the AAVs used (AAV8 or AAV9-PhP.eB). IHCs from $Myo7a^{Sh1/Sh1}$ mice injected with AAV-*Myo7a* had a larger expression of the basolateral K$^+$ current characteristic of mature cells ($I_{K,f}$) compared to $Myo7a^{Sh1/Sh1}$ mice. These

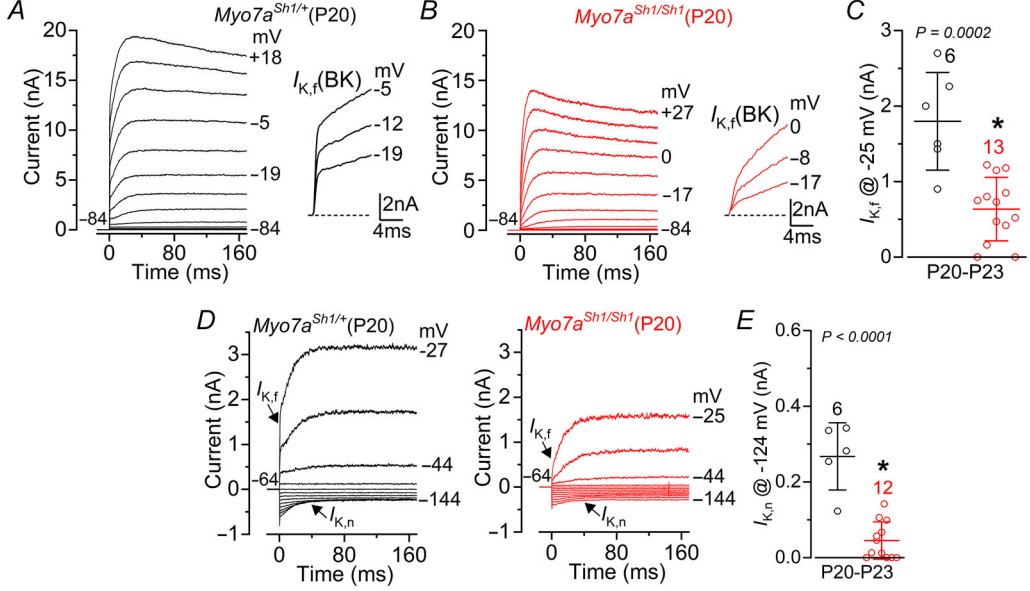

**Figure 4. Mature IHCs from *shaker-1* mice retain an immature-like basolateral membrane current profile**
*A* and *B*, current responses from IHCs of control $Myo7a^{Sh1/+}$ (*A*) and $Myo7a^{Sh1/Sh1}$ (*B*) P20 mice. The fast activation of the BK current ($I_{K,f}$) is better appreciated in the expanded time scale (see insets). *C*, the size of the outward K$^+$ current $I_{K,f}$, which was measured at −25 mV and at 1 ms from the onset of the voltage step (Marcotti et al., 2003). *D*, current responses from IHCs of control $Myo7a^{Sh1/+}$ (left) and $Myo7a^{Sh1/Sh1}$ (right) P20 mice, elicited by using hyperpolarizing and depolarizing voltage steps (10 mV increments) from the holding potential of −64 mV to the various test potentials shown by some of the traces. This protocol is used to emphasize the presence of $I_{K,n}$. *E*, the size of $I_{K,n}$, which was measured as the difference between the peak and steady state of the deactivating inward current at −124 mV from control (black) and $Myo7a^{Sh1/Sh1}$ (red) P20–P23 mice. Data in *C* and *E* are plotted as mean ± SD. Single cell value recordings (open symbols) are plotted behind the average data. Statistical tests shown are obtained using a *t* test. Number of IHCs investigated is shown above the average data points.

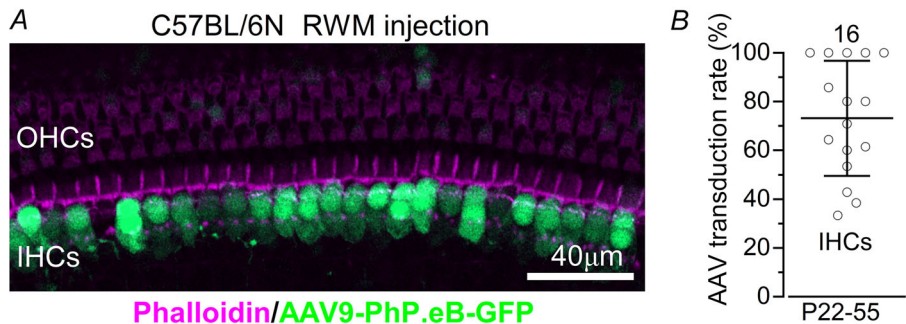

**Figure 5. Hair cell transduction efficiency of AAVs**
*A*, confocal images obtained from the apical coil of the cochlea from wild-type mice (C57BL/6N), which were transduced with AAVs-*GFP* (AAV8-*GFP* or AAV9-PhP.eB-*GFP*) through the round window membrane (RWM: see Methods) at P1–P3. Dissected cochleae were fixed, stained with Texas red phalloidin, and imaged for both phalloidin and GFP. While the large majority of IHCs were GFP positive, only a few OHCs expressed GFP. *B*, viral-transduction rates in apical IHCs were determined from the number of GFP-positive cells normalized by the total hair cells identified with phalloidin in the field of view. Data are plotted as mean ± SD.

changes translated into improved ABR thresholds of up to about 30 dB between the 6 and 30 kHz frequency range. We propose that exogenous gene augmentation using AAV vectors *in vivo* is a suitable strategy to rescue complex morphological and functional defects present in the hair cells of a mouse model of *USH1B* syndrome. Further hearing improvements would require the use of AAVs that also target OHCs such as PHP.B (György et al., 2018; Shubina-Oleinik et al., 2021) and possibly performing *in vivo* transduction during embryonic stages (Bedrosian et al., 2006; Hu et al., 2020; Iranfar et al., 2025), which will allow expression of the exogenous *Myo7a* prior to any morphological hair bundle defects.

## MYO7A is crucial for hair bundle development and maintenance

The precise staircase-like architecture of stereocilia within the hair bundles of both cochlear IHCs and OHCs is established primarily during embryonic and early post-natal stages through a tightly controlled process of elongation and thickening (Vélez-Ortega & Frolenkov, 2019). This sophisticated control over the growth of stereocilia is regulated by several actin-binding proteins and unconventional myosin motors. Since several of these crucial proteins are required during embryonic stages, their absence leads to largely undeveloped hair bundles in cochlear hair cells, resulting in hearing loss (Fang et al., 2015; Krey et al., 2020; Tadenev et al., 2019). One

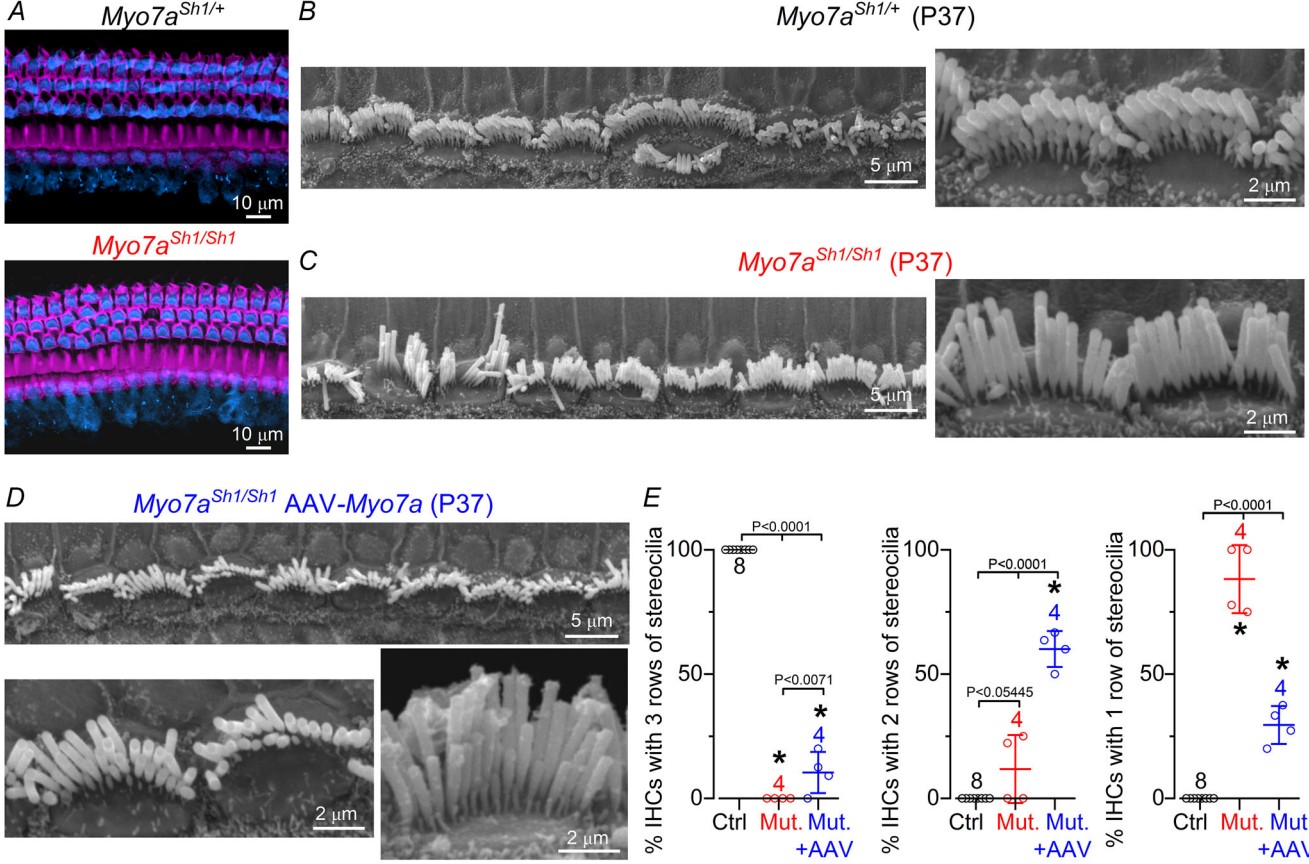

**Figure 6. Improved hair bundle morphology in *Myo7a^{Sh1/Sh1}* mice injected with AAV-*Myo7a***
*A*, maximum intensity projections of confocal z-stack images taken from the apical region of the cochlea in non-injected control (top) and *Myo7a^{Sh1/Sh1}* (bottom) adult mice showing that MYO7A (blue) is expressed in all hair cells. The stereocilia actin core was labelled with phalloidin (magenta). *B–D*, scanning electron microscope images showing the IHC hair bundle structure in the apical coil region of the cochlea of P37 control *Myo7a^{Sh1/+}* (*B*), *Myo7a^{Sh1/Sh1}* (*C*) and *Myo7a^{Sh1/Sh1}* AAV-Myo7a (*D*) mice at different magnifications. *E*, average number of IHCs in the field of view (about 60 μm) showing three rows (left), two rows (middle) or one row (right) of stereocilia. Data are plotted as mean ± SD. Single cell value recordings (open symbols) are plotted behind the average bars. Statistical tests shown are obtained using Tukey's post test (one-way ANOVA). Number of cochleae investigated is shown above the average data points.

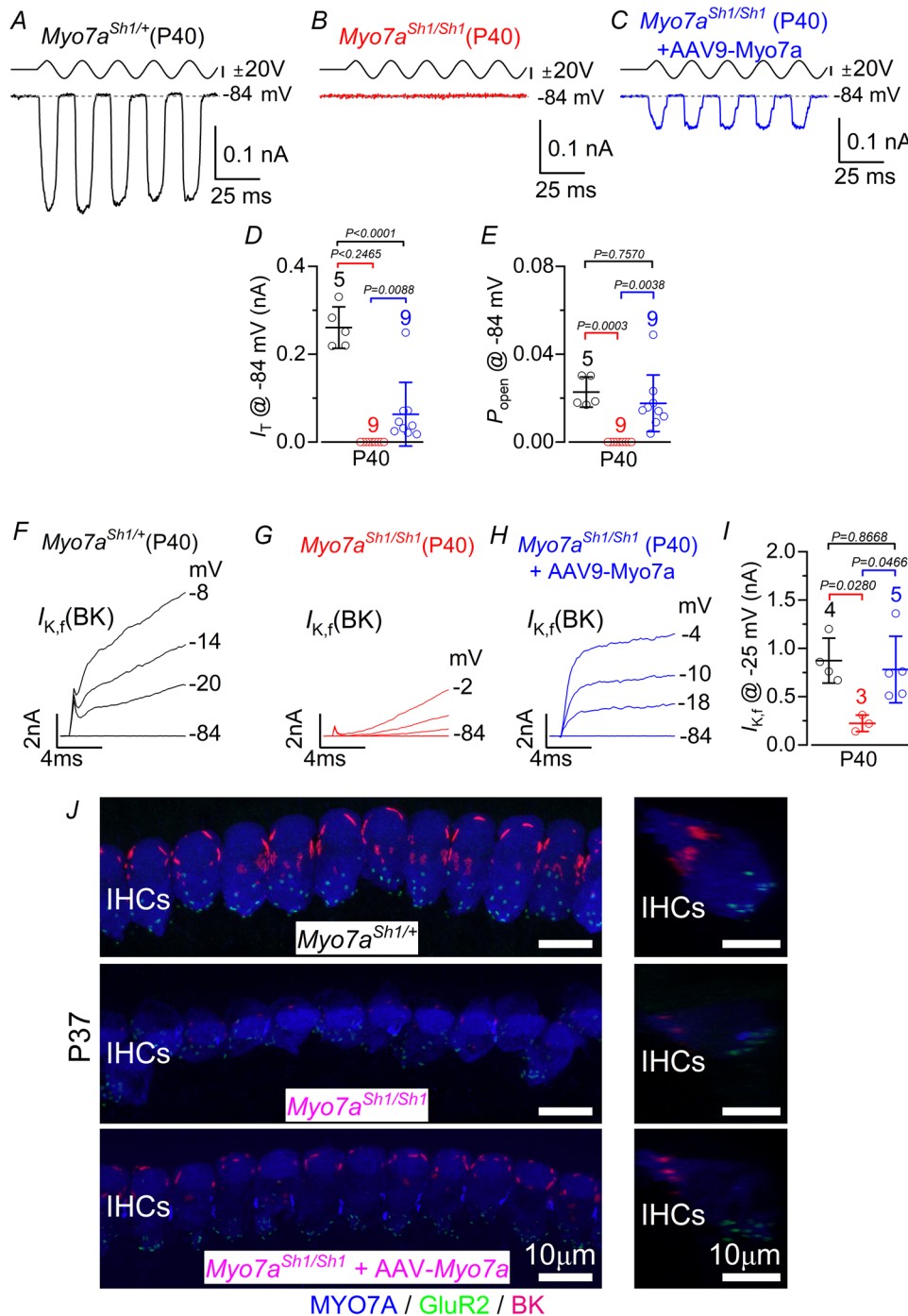

**Figure 7. IHCs from *Myo7a^Sh1* mice transduced with AAV-*Myo7a* delivery recover mechanoelectrical transduction and basolateral membrane potassium currents**

*A–C*, saturating MET currents recorded from apical-coil IHCs of P40 control (*A*), *Myo7a^Sh1/Sh1* (*B*) and *Myo7a^Sh1/Sh1* AAV-Myo7a (*C*) mice. MET currents were elicited in response to 50 Hz sinusoidal force stimuli to the hair bundles at the membrane potential of –84 mV in the presence of 1 mM intracellular EGTA. Driver voltage (DV) stimuli to the fluid jet are shown above the traces (positive DV being excitatory). *D*, maximum size of the MET current recorded from P40 IHCs from control (black), *Myo7a^Sh1/Sh1* (red) and *Myo7a^Sh1/Sh1* AAV-Myo7a (blue) mice at –84 mV. *E*, resting open probability ($P_o$) of the MET channel in P40 IHCs from the three experimental conditions at –84 mV. *F–H*, activation time course of the BK current ($I_{K,f}$) recorded from IHCs of control (*F*), *Myo7a^Sh1/Sh1* (*G*) and *Myo7a^Sh1/Sh1* AAV-Myo7a (*H*) P40 mice. *I*, size of $I_{K,f}$ measured at –25 mV and at 1 ms from the onset of the voltage step. Data in *D*, *E* and *I* are plotted as mean ± SD. Single cell value recordings (open symbols) are plotted behind the average bars. Statistical tests shown are obtained using Tukey's post test (one-way ANOVA). Number of IHCs

investigated is shown above the average data points. *J*, maximum intensity projections of confocal z-stacks taken from the apical cochlear region of control (top), *Myo7a*[Sh1/Sh1] (middle) and *Myo7a*[Sh1/Sh1] *AAV8-Myo7a* (bottom) P40 mice using antibodies against BK (red: located at the IHC neck region), the post-synaptic marker GluR2 (green) and the hair cell marker MYO7A (blue). Right panels show maximum intensity projections of confocal z-stack images of one IHC from the right panels but viewed from the side. Scale bars are 10 μm.

of the key molecules regulating the formation of the stereocilia is MYO7A, which is highly expressed along the entire length of the stereocilia and cell body of both IHCs and OHCs (Hasson et al., 1997; Underhill et al., 2025). Most of the *Myo7a* mutations in the *shaker-1* locus, which target the motor head of the protein, lead to profoundly disorganized hair bundles (e.g. loss of their characteristic shape and staircase structure: Kros et al., 2002; Self et al., 1998). One exception is the original *shaker-1* mutant mouse (*Myo7a*[Sh1]) in which the hair bundles are able to acquire a normal structure but start losing the shortest rows of stereocilia prior to the onset of hearing. How exactly MYO7A promotes the development and maintenance of the stereociliary bundles is still largely unclear, although several roles have been proposed over the years.

MYO7A, like other unconventional myosin motors expressed in the mammalian cochlea (e.g. MYO3A, MYO15A and possibly MYO1), uses its motor activity to deliver key proteins to the stereocilia (Miyoshi et al., 2024; Moreland & Bird, 2022). For example, MYO7A is required for transporting the barbed-end capping protein twinfilin 2 to the stereocilia tip, which has been shown to regulate the elongation of the shorter transducing stereocilia (Peng et al., 2009; Rzadzinska, et al., 2009) that are the primary target of the *shaker-1* mutation. The assembly of the *USH2* protein complex at stereocilia ankle links, which are filaments connecting adjacent stereocilia at their base during development (between P2 and P9: Goodyear et al., 2005), has also been shown to require MYO7A through interaction with the scaffolding protein

PDZD7 (Grati et al., 2012; Michalski, et al., 2007; Morgan et al., 2016; Zou et al., 2017). Abnormal crosslinks in *Myo7a*-deficient mice have been suggested to impact the maintenance of hair-bundle integrity and/or reabsorption of the stereocilia that are not fully linked to the rest of the stereocilia bundle (Hasson et al., 1997). These and other activities carried out by MYO7A, including its interactions with additional stereociliary proteins (Miyoshi et al., 2024; Moreland & Bird, 2022), show that MYO7A plays a key role in the acquisition and maintenance of the structural and functional integrity of the hair bundles, a role that has recently been shown in the mature cochlea (Underhill et al., 2025).

## Hair cell maturation requires a functional MET apparatus prior to the onset of hearing

The functional maturation of cochlear hair cells is initiated by intrinsic genetic programmes that are coordinated by several transcription factors (Pyott et al., 2024). However, these genetic programmes are influenced and guided by $Ca^{2+}$-dependent activity that occurs in the hair cells during a critical period of cochlear development (Ceriani et al., 2019; De Faveri et al., 2025; Johnson et al., 2011; Tritsch & Bergles, 2010). This spontaneous activity has been shown to regulate the remodelling of synapses and ion-channel expression not only in cochlear hair cells (Carlton et al., 2023; Ceriani et al., 2019; Johnson et al., 2007, 2013) but also in cells from other systems (Moody & Bosma, 2005; Zhang & Poo, 2001), probably by modulating gene expression (Dolmetsch et al, 1997).

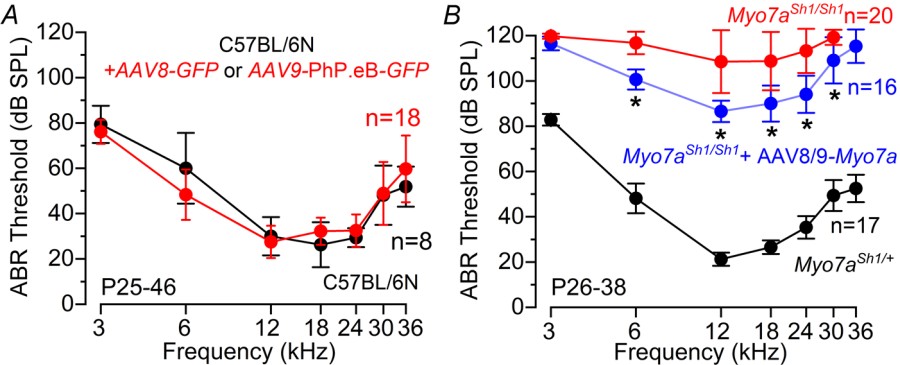

**Figure 8. Improved ABR thresholds in *Myo7a*[Sh1/Sh1] mice injected with AAV-*Myo7a***
*A*, ABR thresholds for frequency-specific pure tone stimulation between 3 and 36 kHz recorded from non-injected (black) and injected (red: surgery and RWM injection of AAVs-*GFP*) mice at P25–P46. *B*, average ABR thresholds for frequency-specific pure tone burst stimuli recorded from control *Myo7a*[Sh1/+] (black), *Myo7a*[Sh1/Sh1] (red) and *Myo7a*[Sh1/Sh1] *AAVs-Myo7a* (blue: AAV8-*GFP* or AAV9-PhP.eB-*GFP*) mice at P26–38. The number of mice tested for each experimental conditions are shown next to the data (mean ± SD).

Although $Ca^{2+}$-dependent signals in the developing cochlea occur in the absence of external stimuli (De Faveri et al., 2025), they are continuously modulated by several physiological mechanisms, including the activity of the MET channel (Ceriani et al., 2025). This is because from about P7 onwards, a time when the MET apparatus of hair cells is already functional (Lelli et al., 2009; Waguespack et al., 2007), the endocochlear potential starts to build up (Li et al., 2020). This generates the driving force for a sustained depolarizing MET current, which is crucial for keeping the resting membrane potential of the hair cells close to the activation threshold of $Ca^{2+}$-dependent action potentials (De Faveri et al., 2025; Johnson et al., 2012). Disruption of the MET current, for example in mice carrying mutations in genes targeting either hair bundle morphology (e.g. *Eps8*) or the opening or function of the MET channels (e.g. *Pcdh15*, *Ush1c*, *Tmc1*), alters the $Ca^{2+}$ signalling dynamics in developing hair cells, which has been shown to prevent maturation of the hair cell basolateral membrane profile. This lack of maturation, which compromises the ability of hair cells to process acoustic information, includes a failure to upregulate key ion channels characteristic of mature hair cells, such as KCNQ4 and BK channels (Corns et al., 2018; Marcotti et al., 2006; Zampini et al., 2011), and the refinement of the afferent (Lee et al., 2021; Sun et al., 2018) and efferent synapses (Corns et al., 2018). Therefore, it is likely that the abnormal basolateral profile of mature hair cells from $Myo7a^{Sh1}$ mice is a consequence of the altered MET apparatus in developing hair cells, rather than from a direct role of MYO7A in targeting proteins to the cell membrane.

### *In vivo* delivery of AAV-*Myo7a* partially rescues IHC function and hearing loss in *shaker-1* mice

Recent advances in AAV-mediated gene therapy approaches targeting hair cell genetic defects responsible for deafness have been instrumental in advancing several preclinical trials using murine models of hereditary deafness (Amariutei et al., 2023; Petit et al., 2023). Currently, the most successful studies have implicated genes involved in exocytosis (*Vglut3*: Akil et al., 2012; *Otof*: Akil et al., 2019; Al-Moyed et al., 2019) and mechanoelectrical transduction (*Tmc1*: Askew et al., 2015; Nist-Lund et al., 2019). However, the rescue of deafness genes crucial for the initial growth of the hair cell stereociliary bundles such as *Eps8* (Jeng et al., 2022), *Clarin-1* (Dulon et al., 2018) and *Sans* (Emptoz et al., 2017) have shown limited success. Similarly, a recent attempt to restore hearing using gene-based therapy in mice with a nonsense mutation in *Myo7a* ($Myo7a^{4626SB/4626SB}$), which is a model for *USH1B*, did not result in detectable hearing improvements (Lau et al., 2023). In contrast, we found that injecting AAV-*Myo7a* into the cochlea of $Myo7a^{Sh1/Sh1}$

mice led to partial recovery of the hair cell hair bundle structure, restoration of basolateral membrane physiology in IHCs and an improvement in auditory thresholds (up to about 30 dB). The higher degree of hearing recovery in $Myo7a^{Sh1/Sh1}$ compared to $Myo7a^{4626SB/4626SB}$ mice is likely to be due to the latter having extensive hair bundle disruption from early postnatal stages (Holme & Steel, 2002). This further supports the idea that the effectiveness of AAV gene-based therapies for congenital deafness is dictated largely by the level of morphological damage present at the time of treatment. For genes involved in the formation of the sensory epithelium, such as most of those implicated in *USHER* syndromes, there is a narrow therapeutic window early in development, necessitating *in utero* AAV-mediated gene delivery, as previously demonstrated for other genes (Bedrosian et al., 2006; Hu et al., 2020). An alternative possibility would be to extend the therapeutic window by delaying the progression of the morphological dysfunction.

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

## Additional information

### Data availability statement

The data that support the findings of this study are available from the corresponding authors.

### Competing interests

The authors declare no conflicts of interest.

### Author contributions

All authors helped with the collection and analysis of the data. W.M. conceived and coordinated the study. All authors approved the final version of the manuscript. All authors agree to be accountable for all aspects of the work in ensuring that questions related to the accuracy or integrity of any part of the work are appropriately investigated and resolved. All persons designated as authors qualify for authorship, and all those who qualify for authorship are listed.

### Funding

This work was supported by the French National Research Agency, funding the France 2030 program entitled RHU AUDINNOVE (ANR-18-RHUS-0007 to S.S.). This work was also supported by grants from the "Fondation pour l'Audition" (FPA IDA08 to S.S.), BBSRC (BB/Z516685/1) and Wellcome Trust (224326/Z/21/Z) to W.M.; BBSRC (BB/X000567/1) to S.L.J.; Wellcome Trust (300350/Z/23/Z) to A.J.C.; BBSRC (BB/Z514743/1) to J.-Y.J. A.O.'C. was supported by a PhD studentship from the RNID (S56) Partnership to W.M. A.E.A. was supported by a PhD studentship from the Sheffield Neuroscience Institute to W.M. A.U. was supported by a PhD studentship from the MRC DiMeN Doctoral training Partnership to W.M. For the purpose of Open Access, the author has applied a CC BY public copyright licence to any Author Accepted Manuscript version arising from this submission.

### Acknowledgements

The authors thank Steven M. Barnes and Matthew A. Loczki at the University of Sheffield for their assistance with the mouse husbandry, and Catherine Gennery and Niovi Voulgari for their genotyping work.

### Keywords

cochlea, deafness, gene-based therapy, hair cell, ion channels, mechanoelectrical transduction, myosin motor, ribbon synapses

### Supporting information

Additional supporting information can be found online in the Supporting Information section at the end of the HTML view of the article. Supporting information files available:

**Peer Review History**

