## [Peer Review History · The Journal of Physiology]

AAV-based rescue of *Myo7a* expression restores hair-cell function and improves hearing thresholds in a *USH1B* mouse strain

Ana E Amariutei, Samuel Webb, Adam Carlton, Andrew O'Connor, Anna Underhill, Jing-Yi Jeng, Sarah A Hool, Alice Zanella, Matthew Hool, Marie-José LECOMTE, Stuart Leigh Johnson, Saaid Safieddine, and Walter Marcotti
DOI: 10.1113/JP289526

Corresponding author(s): Walter Marcotti (w.marcotti@sheffield.ac.uk)

The following individual(s) involved in review of this submission have agreed to reveal their identity: Keiko Hirose (Referee #2)

Review Timeline:

Submission Date:	18-Jun-2025
Editorial Decision:	29-Jul-2025
Revision Received:	21-Aug-2025
Accepted:	28-Aug-2025

Senior Editor: Vaughan Macefield

Reviewing Editor: Conny Kopp-Scheinflug

Transaction Report:

Dear Dr Marcotti,

Re: JP-RP-2025-289526 "**AAV-based rescue of *Myo7a* expression restores hair-cell function and improves hearing in a *USH1B* mouse strain**" by Ana E Amariutei, Samuel Webb, Adam Carlton, Andrew O'Connor, Anna Underhill, Jing-Yi Jeng, Sarah A Hool, Alice Zanella, Matthew Hool, Marie-José LECOMTE, Stuart Leigh Johnson, Saaid Safieddine, and Walter Marcotti

Thank you for submitting your manuscript to The Journal of Physiology. It has been assessed by a Reviewing Editor and by 2 expert referees and we are pleased to tell you that it is acceptable for publication following satisfactory revision.

REVISION CHECKLIST:

Please upload two versions of your manuscript text: one with all relevant changes highlighted and one clean version with no changes tracked. The manuscript file should include all tables and figure legends, but each figure/graph should be uploaded as separate, high-resolution files. The journal is now integrated with Wiley's Image Checking service. For further details,

see: <https://www.wiley.com/en-us/network/publishing/research-publishing/trending-stories/upholding-image-integrity-wileys-image-screening-service>

We look forward to receiving your revised submission.

Yours sincerely,

Vaughan Macefield
Senior Editor
The Journal of Physiology

REQUIRED ITEMS

- Author photo and profile. First or joint first authors are asked to provide a short biography (no more than 100 words for one author or 150 words in total for joint first authors) and a portrait photograph. These should be uploaded and clearly labelled together in a Word document with the revised version of the manuscript. See Information for Authors for further details.

- You must start the Methods section with a paragraph headed Ethical approval (https://jp.msubmit.net/cgi-bin/main.plex?form_type=display_requirements#methods).

Research must comply with The Journal's policies regarding animal experiments (<https://physoc.onlinelibrary.wiley.com/hub/animal-experiments>) and adherence to these policies must be stated in the manuscript.

Authors should confirm in their Methods section that their experiments were carried out according to the guidelines laid down by their institution's animal welfare committee, including an ethics approval reference number. The Methods section must contain a statement about access to food, water and housing, details of the anaesthetic regime: anaesthetic used, dose and route of administration, and method of killing the experimental animals.

- Please upload separate high-quality figure files via the submission form.

- We invite you to include a Translational Perspective paragraph in your manuscript. This should be included in the main body of the manuscript after the Acknowledgements. It should describe the wider translational implications of the work, in plain English, for a broad scientific audience. Please use the following guidelines to prepare a Translational Perspective of your paper: https://jp.msubmit.net/cgi-bin/main.plex?form_type=display_requirements#authortranspersp. The Translational Perspective should not exceed 250 words in total and should be presented as a single paragraph. Abbreviations and technical terms must be defined as briefly and simply as possible the first time they are used, unless they are generally/easily understood, e.g. ECG, HIV/AIDS, K⁺ channel. Use language that can be understood by scientists or clinicians with a general knowledge of the topic addressed. Ensure the paragraph includes the hypothesis tested in the paper and accurately reflects the findings of the paper and the implications for future research. Please state the word count of the Translational Perspective paragraph.

- Please include an Abstract Figure file, as well as the Figure Legend text within the main article file. The Abstract Figure is a piece of artwork designed to give readers an immediate understanding of the research and should summarise the main conclusions. If possible, the image should be easily 'readable' from left to right or top to bottom. It should show the physiological relevance of the manuscript so readers can assess the importance and content of its findings. Abstract Figures should not merely recapitulate other figures in the manuscript. Please try to keep the diagram as simple as possible and without superfluous information that may distract from the main conclusion(s). Abstract Figures must be provided by authors no later than the revised manuscript stage and should be uploaded as a separate file during online submission labelled as

File Type 'Abstract Figure'. Please also ensure that you include the figure legend in the main article file. All Abstract Figures should be created using BioRender. Authors should use The Journal's premium BioRender account to export high-resolution images. Details on how to use and access the premium account are included as part of this email.

EDITOR COMMENTS

Reviewing Editor:

Your manuscript has been reviewed by two specialists in the field and received quite enthusiastic reviews. Please attend to the minor suggestions made by the reviewers before submitting the final version. Please also don't forget to state the animals access to food and water as JP policy asks for.

Please also see 'Required Items' above.

Senior Editor:

Thank you for submitting your manuscript to The Journal of Physiology. As you will see from the Reviewing Editor's comments, and those of the two independent reviewers, we believe your manuscript will contribute importantly to the field. However, before we can accept it for publication, please attend to the comments raised by the Reviewing Editor and reviewers. We look forward to receiving your revised manuscript in due course.

REFEREE COMMENTS

Referee #1:

Given that gene therapy clinical trials in children with pathogenic variants in the OTOF gene have demonstrated significant improvements in hearing thresholds, it becomes imperative to rescue hearing for as many deafness genes as possible. There are over 200 deafness-related genes found to date, with gene replacement using AAV done for at least 16 of these genes in mice. In this manuscript, the authors chose to attempt to rescue deafness in the shaker-1 mouse model for MYO7A human deafness. Due to the large size of Myo7a, two AAVs were prepared, each with a portion of the gene. The authors performed a thorough analysis to demonstrate partial functional recovery of hearing. Morphology was examined using SEM, functional tests using MET currents (mechanoelectrical transduction and basolateral membrane potassium currents), and hearing tests using ABR.

Although the authors show improvements in ABR threshold, the thresholds they are using are very high. Therefore, "partially restores hearing" is overstated. It would be better to state "improvements in ABR thresholds"

Myo7a affects outer hair cells as well, therefore OHC should have been a target as well. Please address why you did not use AAVs that also target OHC (for example AAV9-PHP.B). "Exogenous delivery of MYO7A using dual-AAV vectors in Myo7aSh1/Sh1 P0-P2 pups in vivo was able to partially restore the staircase structure of the hair bundles and the MET current in adult IHCs, which were the main target of the AAVs used (AAV8 or AAV9-PhP.eB)".

Line 356, do not use contractions

Referee #2:

This manuscript describes in detail the morphology and physiology associated with the shaker-1 spontaneous mouse

mutant that carries two mutations in the gene for myosin 7A. The characterization of this mouse is beautifully described and laid out, and the attempt to correct the hearing loss with AAV8 and 9 is a clearly and well rationalized. The data are clearly presented, the photomicrographs are easily interpreted and the graphs are clean and easy to read. The experiments were repeated with sufficient repetition for the data to be persuasive.

The resultant improvement in hearing threshold is disappointing, in that there remains a significant hearing loss, and this is with gene therapy delivered as soon as the mouse is born. While there may be limited merit in attempting to use AAV gene therapy for Myo7Ash/sh in humans, this study clearly demonstrates what it is capable of achieving in mice. The fact that the result does not inspire translation to humans does not detract from the importance of the findings. Further areas to explore would include trialing other vectors that might transduce outer hair cells in addition to inner hair cells, including a fluorescent tag to allow visualization of which. and how many hair cells have been infected, and demonstrating if in fact MET channels are reduced in the short stereocilia.

END OF COMMENTS

JP-RP-2025-289526

AAV-based rescue of Myo7a expression restores hair-cell function and improves hearing in a USH1B mouse strain

We thank the Reviewers for their comments, which we have addressed in the revised manuscript. Line numbers refer to: Amariutei et al 2025_Revised_Changes Highlighted.pdf

Reviewing Editor:

Your manuscript has been reviewed by two specialists in the field and received quite enthusiastic reviews. Please attend to the minor suggestions made by the reviewers before submitting the final version. Please also don't forget to state the animals access to food and water as JP policy asks for.

Thank you.

Please also see 'Required Items'

- Author photo and profile.

Included.

- You must start the Methods section with a paragraph headed Ethical approval

Done (ln. 120).

- Research must comply with The Journal's policies regarding animal experiments (<https://physoc.onlinelibrary.wiley.com/hub/animal-experiments>) and adherence to these policies must be stated in the manuscript.

Added (ln. 139-140).

- Authors should confirm in their Methods section that their experiments were carried out according to the guidelines laid down by their institution's animal welfare committee, including an ethics approval reference number.

Yes, already included in the submitted version (ln. 121-123).

- The Methods section must contain a statement about access to food, water and housing, details of the anaesthetic regime: anaesthetic used, dose and route of administration, and method of killing the experimental animals.

The missing information about food and water availability has been included in the revised manuscript (ln. 123-124).

- Please include an Abstract Figure file, as well as the Figure Legend text within the main article file.

Done (ln. 832-839)

Senior Editor:

Thank you for submitting your manuscript to The Journal of Physiology. As you will see from the Reviewing Editor's comments, and those of the two independent reviewers, we believe your manuscript will contribute importantly to the field. However, before we can accept it for publication, please attend to the comments raised by the Reviewing Editor and reviewers. We look forward to receiving your revised manuscript in due course.

Thank you

Reviewer 1

Given that gene therapy clinical trials in children with pathogenic variants in the OTOF gene have demonstrated significant improvements in hearing thresholds, it becomes imperative to rescue hearing for as many deafness genes as possible. There are over 200 deafness-related genes found to date, with gene replacement using AAV done for at least 16 of these genes in mice. In this manuscript, the authors chose to attempt to rescue deafness in the shaker-1 mouse model for MYO7A human deafness. Due to the large size of Myo7a, two AAVs were prepared, each with a portion of the gene. The authors performed a thorough analysis to demonstrate partial functional recovery of hearing. Morphology was examined using SEM, functional tests using MET currents (mechanoelectrical transduction and basolateral membrane potassium currents), and hearing tests using ABR.

Thank you

Although the authors show improvements in ABR threshold, the thresholds they are using are very high. Therefore, "partially restores hearing" is overstated. It would be better to state "improvements in ABR thresholds"

We have changed the text as indicated throughout the manuscript when referring to "partial restore hearing".

Myo7a affects outer hair cells as well, therefore OHC should have been a target as well. Please address why you did not use AAVs that also target OHC (for example AAV9-PHP.B). "Exogenous delivery of MYO7A using dual-AAV vectors in Myo7aSh1/Sh1 P0-P2 pups in vivo was able to partially restore the staircase structure of the hair bundles and the MET current in adult IHCs, which were the main target of the AAVs used (AAV8 or AAV9-PhP.eB)".

Unfortunately, there is not an ideal AAV that is able to target both hair cells with high efficiency, at least in our hands. Our choice of AAV8 and AAV9-PHP.eB vectors was guided by their greater IHC tropism and transduction efficiency in neonatal and early postnatal stages, which we confirmed in preliminary experiments.

While AAV9-PHP.B has shown broader tropism including OHCs, its efficiency and reproducibility across mouse strains and ages can be very variable. Moreover, our primary objective was to establish proof of concept for MYO7A replacement in IHCs using a dual-AAV approach. We agree that future studies targeting OHC, potentially with other capsids such as PHP.B, will be valuable to further improve functional outcomes and fully restore cochlear function in the Myo7a-deficient model. We now mention this in the revised text (ln. 414-420).

Line 356, do not use contractions

Corrected

Reviewer 2

This manuscript describes in detail the morphology and physiology associated with the shaker-1 spontaneous mouse mutant that carries two mutations in the gene for myosin 7A. The characterization of this mouse is beautifully described and laid out, and the attempt to correct the hearing loss with AAV8 and 9 is clearly and well rationalized. The data are clearly presented, the photomicrographs are easily interpreted and the graphs are clean and easy to read. The experiments were repeated with sufficient repetition for the data to be persuasive.

Thank you

The resultant improvement in hearing threshold is disappointing, in that there remains a significant hearing loss, and this is with gene therapy delivered as soon as the mouse is born. While there may

be limited merit in attempting to use AAV gene therapy for Myo7Ash/sh in humans, this study clearly demonstrates what it is capable of achieving in mice. The fact that the result does not inspire translation to humans does not detract from the importance of the findings. Further areas to explore would include trialing other vectors that might transduce outer hair cells in addition to inner hair cells, including a fluorescent tag to allow visualization of which, and how many hair cells have been infected, and demonstrating if in fact MET channels are reduced in the short stereocilia.

We completely agree, and we have emphasized this in the revised manuscript (ln. 414-420). This is of course a major issue in the field that needs addressing if we were to improve the outcome of most mutations effecting both hair cell types.

Dear Professor Marcotti,

Re: JP-RP-2025-289526R1 "**AAV-based rescue of *Myo7a* expression restores hair-cell function and improves hearing thresholds in a *USH1B* mouse strain**" by Ana E Amariutei, Samuel Webb, Adam Carlton, Andrew O'Connor, Anna Underhill, Jing-Yi Jeng, Sarah A Hool, Alice Zanella, Matthew Hool, Marie-José LECOMTE, Stuart Leigh Johnson, Saaid Safieddine, and Walter Marcotti

We are pleased to tell you that your paper has been accepted for publication in The Journal of Physiology.

Yours sincerely,

Vaughan Macefield
Senior Editor
The Journal of Physiology

If you would like to receive our 'Research Roundup', a monthly newsletter highlighting the cutting-edge research published in The Physiological Society's family of journals (The Journal of Physiology, Experimental Physiology, Physiological Reports, The Journal of Nutritional Physiology and The Journal of Precision Medicine: Health and Disease), please click this link, fill in your name and email address and select 'Research Roundup':
<https://www.physoc.org/journals-and-media/membernews>

- You can help your research get the attention it deserves! Check out Wiley's free Promotion Guide for best-practice recommendations for promoting your work at: www.wileyauthors.com/eoo/guide. You can learn more about Wiley Editing Services which offers professional video, design, and writing services to create shareable video abstracts, infographics, conference posters, lay summaries, and research news stories for your research at: www.wileyauthors.com/eoo/promotion.

EDITOR COMMENTS

Reviewing Editor:

Great paper - congratulations.

Senior Editor:

Thank you for attending to these remaining issues. I am pleased to report that your manuscript is now considered acceptable for publication in The Journal of Physiology.